# Engineering fungal *de novo* fatty acid synthesis for short chain fatty acid production

Jan Gajewski[1,*], Renata Pavlovic[2,*], Manuel Fischer[1], Eckhard Boles[2] & Martin Grininger[1]

Fatty acids (FAs) are considered strategically important platform compounds that can be accessed by sustainable microbial approaches. Here we report the reprogramming of chain-length control of *Saccharomyces cerevisiae* fatty acid synthase (FAS). Aiming for short-chain FAs (SCFAs) producing baker's yeast, we perform a highly rational and minimally invasive protein engineering approach that leaves the molecular mechanisms of FASs unchanged. Finally, we identify five mutations that can turn baker's yeast into a SCFA producing system. Without any further pathway engineering, we achieve yields in extra-cellular concentrations of SCFAs, mainly hexanoic acid ($C_6$-FA) and octanoic acid ($C_8$-FA), of 464 mg l$^{-1}$ in total. Furthermore, we succeed in the specific production of $C_6$- or $C_8$-FA in extracellular concentrations of 72 and 245 mg l$^{-1}$, respectively. The presented technology is applicable far beyond baker's yeast, and can be plugged into essentially all currently available FA overproducing microorganisms.

[1] Institute of Organic Chemistry and Chemical Biology, Buchmann Institute for Molecular Life Sciences, Cluster of Excellence 'Macromolecular Complexes', Goethe University Frankfurt, Max-von-Laue-Strasse 15, 60438 Frankfurt, Germany. [2] Institute of Molecular Biosciences, Department of Biological Sciences, Goethe University Frankfurt, Max-von-Laue-Strasse 9, 60438 Frankfurt, Germany. * These authors contributed equally to this work. Correspondence and requests for materials should be addressed to E.B. (email: e.boles@bio.uni-frankfurt.de) or to M.G. (email: grininger@chemie.uni-frankfurt.de).

Fatty acids (FAs) are important molecules serving several purposes in living entities; that is, they are building blocks of membranes, signalling molecules and energy depots[1,2]. In the production of fine chemicals from renewable resources, FAs can play a crucial role as platform compounds[3,4]. Accordingly, the production of FAs in bacteria and fungi has been investigated extensively, constantly overcoming technical boundaries. In *Saccharomyces cerevisiae*, titres of $400 \, mg \, l^{-1}$ free FAs[3], and more recently of $2.2 \, g \, l^{-1}$ (ref. 5) and even up to $10.4 \, g \, l^{-1}$ in a glucose limited fed-batch cultivation[6] have been achieved (mainly long FAs, from $C_{12}$ to $C_{18}$ in saturated and monounsaturated form). In *E. coli*, titres of up to $5.2 \, g \, l^{-1}$ (predominantly $C_{14}$ and $C_{16:1}$; in $C_{n:m}$ the $n$ represents the number of carbons in the FA, $m$ stands for the desaturated bonds) were reported[7]. The native FA product spectrum of living organisms is covering mainly $C_{14}$- to $C_{18}$-FA (ref. 2), which restricts microbial production to long FAs. As FAs in the range of $C_4$ to $C_{12}$ are technologically relevant, for example as precursors for biofuels in the petrol range[8], substantial efforts have been invested for harnessing bacterial and fungal fatty acid synthases (FASs) for the microbial production of short-chain fatty acids (SCFAs). Recent approaches employing thioesterases with known specificities for short-chain products have succeeded in the production of SCFAs in *S. cerevisiae* with titres of up to $111 \, mg \, l^{-1}$ (ref. 5), and also have achieved the production of FA derived alkanes in *E. coli*[4]. Both approaches were based on interfering in native synthesis with a short acyl-ACP-specific thioesterase, which was either provided as separate protein or as a domain fused to FAS. Another more recent approach succeeded in producing $C_{12}$- and $C_{14}$-FA with fungal FAS, and short acyl-ACP thioesterases similarly provided either as separate proteins or as domains inserted in the protein complex[9].

We recently started the project of rewriting chain-length control in fungal *de novo* fatty acid synthesis, encouraged by the idea of performing a rational engineering approach that is minimally invasive to the protein's molecular mechanisms. The fungal FAS is a highly efficient FA producing machinery that runs synthesis at higher turnover numbers than other type I FAS systems (18.2, 2.0 and 3.4 cycles per second; values per set of active sites calculated from specific activities reported for *S. cerevisiae*[10], *Corynebacterium ammoniagenes*[11] and chicken FAS[12], respectively). The synthetic advantage of the fungal FAS lies in its highly developed architecture, in which the catalytic domains are embedded in an extensive scaffolding matrix, and substrate shuttling by the acyl carrier protein (ACP) is subtly balanced between electrostatic steering and molecular crowding properties[13–15] (Fig. 1a). In focusing on engineering strategies that address substrate specificities, but leave the overall structural properties of the fungal FAS unchanged, we aimed at steering the systems towards the premature release of the not yet completely elongated acyl-chain, while conserving the intrinsic efficiency of fungal *de novo* FA synthesis (Fig. 1b). In our approach, we specifically took advantage of the tight coverage of *S. cerevisiae* FAS with structural and functional data. The 2.6 MDa *S. cerevisiae* protein is the best studied fungal FAS, documented for example by a wealth of X-ray structural data at a resolution of up to $3.1 \, Å$ (refs 16–21). Finally, we show that a rationally engineered *S. cerevisiae* FAS, manipulated by up to only five active site mutations, and placed in a yeast wild-type background without any metabolic pathway engineering, is extremely efficient in SCFA production. Strains carrying such engineered FASs give access to specific production of hexanoic acid ($C_6$-FA; titres of $71.5 \, mg \, l^{-1}$ corresponding to 78.2% of the detected SCFAs ($C_6$ to $C_{12}$) in that strain) and of octanoic acid ($C_8$-FA; $245.0 \, mg \, l^{-1}$; 85.9%), and allow maximum titres of extracellular free $C_6$- to $C_{12}$-FA of $464.4 \, mg \, l^{-1}$. In addition,

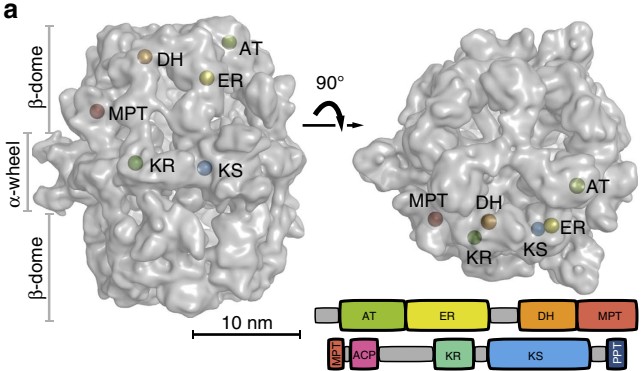

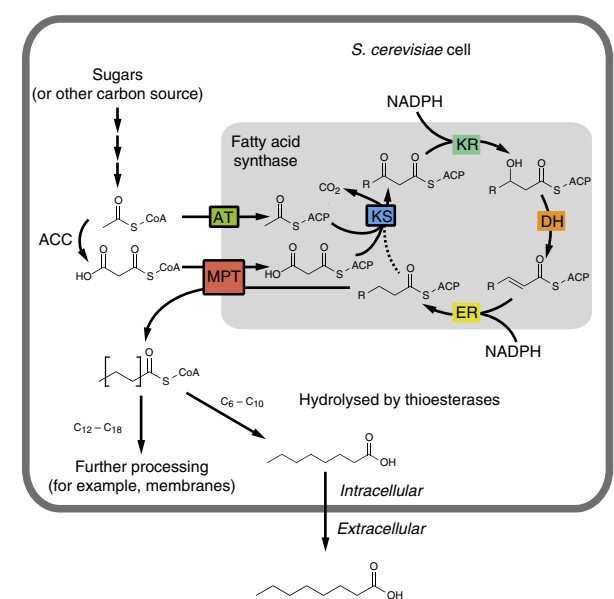

**Figure 1 | Fatty acid synthesis in *S. cerevisiae*.** (**a**) Two polypeptides, the 207 kDa α-chain and the 229 kDa β-chain, assemble to a barrel-shaped $\alpha_6\beta_6$-heterododecameric complex that encloses two reaction chambers, each formed by a β-dome and the α-wheel. The β-chain harbours the domains acetyl transferase (AT), enoyl reductase (ER), dehydratase (DH) and malonyl/palmitoyl transferase (MPT), and the α-chain the domains acyl carrier protein (ACP), ketoacyl reductase (KR), ketoacyl synthase (KS) and the phosphopantetheine transferase (PPT). The barrel-shaped structure of *S. cerevisiae* FAS is shown with one set of active centres indicated (based on protein data bank (PDB) code 3HMJ (ref. 50)). (**b**) In the FA cycle, the KS domain is responsible for the condensation of an acyl substrate (either the starter molecule acetyl or an ACP-bound elongated acyl-chain) with malonyl. The resulting β-ketoacyl intermediate is then processed in a series of reaction steps, performed by the domains KR, DH and ER, to a fully reduced acyl-chain. The resulting acyl-chain serves as a starter for the next cycle. The process is repeated until the final product is cleaved off. Fungal FASs carry ACP domains that are double tethered to the FAS scaffold by unstructured linkers, and shuttle covalently bound substrates and intermediates to the active sites. FAS engineering aimed at producing SCFAs, instead of its native product, typically $C_{16}$- or $C_{18}$-CoA. *S. cerevisiae* FAS naturally produces CoA esters. Free SCFAs, appearing as hydrolysed by unspecific TEs (here shown exemplarily for $C_8$), are then exported to the medium, from which they have been extracted for analysis. The fatty acid cycle is shown in a schematic manner and incomplete in the presentation of side products.

this approach gives access to short-chain acyl-CoA esters, as the immediate products of fungal FA synthesis (Fig. 1b). This is highly advantageous when the desired product portfolio should

extend towards short-chain aldehydes, alcohols and alkanes, as acyl-CoAs can then directly be processed without prior (re)activation of free FAs[4,6,9]. We identify three thioesterases, responsible for the hydrolysis of short-chain acyl-CoA esters in our strains. Thioesterase knockout strains hold out the prospect of efficiently channelling the short-chain acyl-CoA products into such downstream derivatization processes (Fig. 1b).

## Results

**Outline of approach.** For manipulating chain-length regulation in the *de novo* production of SCFAs, key active sites of the condensation domain (KS) and the transferase domains (MPT and AT) were rationally engineered. The other domains (KR, DH and ER), which presumably have no direct influence on chain-length regulation, were not mutated (Fig. 1b). For the evaluation of FAS engineering, a *Δfas1 Δfas2* strain was complemented with vectors carrying α- and β-chain encoding sequences *FAS2* and *FAS1*, respectively. The impact of mutations was generally judged with *FAS2* and *FAS1* under control of their natural promoters and terminators[22]. SCFAs are exported and accumulate in the culture medium[23], where they have been characterized after extraction with gas chromatographic methods (for more information, see Methods).

**Engineering of the KS-mediated elongation step.** The KS domain was considered as the most promising target for rewriting chain-length regulation, as accounting for the elongation of the growing acyl-chain (Fig. 1b). The condensation reaction is performed in a ping-pong mechanism, in which the acyl-intermediate is loaded into the KS binding channel, where it is covalently bound to the active cysteine C1305 (ping step)[24]. Our engineering approach was mainly informed by sequence alignments and structural data, which have been collected on KS domains[19,25]. We mainly focused on M1251, which is protruding into the KS binding channel and dividing the binding channel into an outer and inner volume. On acyl binding, the methionine rotates and gives access to the inner cavity (Fig. 2a). Residues at this position in *S. cerevisiae* FAS and in a homologous protein have been suggested to act as gate keepers that prevent short-chain acyl intermediates from KS-mediated elongation, steering them towards release[19,25]. To build on such an effect, we started KS engineering by introducing the mutation G1250S. *S. cerevisiae* strains with this mutation were characterized as producing higher levels of $C_6$-FA and its ester derivatives[26], presumably by restricting the conformational flexibility of the neighbouring M1251. As a second position in the KS domain, we directly mutated M1251 with the intention to increase the steric barrier (Fig. 2a). In comparison to the wild type, the G1250S strain showed a nine-fold increase in $C_6$-FA production (15.3 mg l$^{-1}$). The G1250S-M1251W double-mutated *S. cerevisiae* strain showed further increased levels of $C_6$-FA (with 19.9 mg l$^{-1}$, 12-fold increase over wild type) and $C_8$-FA (32.7 mg l$^{-1}$, 56-fold increase over wild type; Fig. 2b, Supplementary Fig. 1 and Supplementary Table 1). A third promising KS mutation was identified in sequence alignments with FASs of reported $C_6$-FA producers, such as *Aspergillus nidulans*, *Aspergillus parasiticus* and *Aspergillus flavus*[27,28]. The residue F1279 is protruding into the KS binding channel from the opposite side of G1250S-M1251W (Fig. 2a). Introducing F1279Y into a G1250S-mutated strain strongly affected growth, and FA output spectra were hardly interpretable in chain-length distribution. Beneficial effects of F1279Y were observed in combination with transferase active site mutations, as reported below (Fig. 2b).

**Engineering of FAS-loading and -unloading.** In fungal FAS, the processes of loading of malonyl and unloading of the fully elongated acyl-chain are performed by the MPT domain, in which all substrates are transferred via the same active serine S1808 (ref. 29) (Fig. 1b). As the MPT domain has been shown to broadly accept acyl chains of various chain lengths[30], we engineered MPT for reduced malonyl affinity. As a rationale behind this mutation, a disfavoured malonyl uptake was anticipated to favour the competing release of incompletely elongated short acyl chains. Guided by mutational analysis of homologous proteins[31,32], a R1834K mutation was installed to lower the affinity of malonyl by weakening the interaction with the carboxy group (Fig. 2c). The R1834K mutation led to extracellular SCFA (mostly $C_8$) levels of 100 mg l$^{-1}$ that correspond to a 23-fold increase over wild-type FAS (Fig. 2b).

Fungal FASs catalyse loading of acetyl via the AT domain (Fig. 1b). Previous studies on related transferases reported on elevated acetyl loading after introduction of an arginine to alanine active site mutation[31,32]. We expected that the corresponding I306A mutation is beneficial for SCFA synthesis (particularly in combination with decreased malonyl loading by the R1834K mutation), as elevating acetyl/malonyl ratios can lead to a pronounced priming of FA synthesis[33,34]. Moreover, we considered the I306A mutation as interesting, because of the reported additional effect of broadening substrate specificities of transferases towards short acyl chains[31,32] (Fig. 2c). The product spectra of *S. cerevisiae* carrying the single AT mutation I306A showed no significant effect on SCFA yield. However, in combination with other mutations, I306A-mutated AT drastically changed product distributions (Fig. 2b). In Fig. 2d, putative fluxes in FA synthesis are depicted that are underlying FA production by the respective yeast strains.

**Combinations of mutations.** Mutations were also tested in several combinations, leading to increased amounts of overall SCFAs as well as to a highly selective production of specific SCFAs (Fig. 2b). FA output spectra of strains carrying a multiple mutated FAS generally implied a complex and partly non-intuitive behaviour of individual mutational effects. This is, for example, visible on the effect of the I306A mutation on the FA spectrum. The strain carrying solely the I306A mutation did not show an increased yield in SCFAs compared to the wild-type strain, while the insertion of the I306A mutation into the background of a KS double-mutated strain (yielding a I306A-G1250S-M1250W-mutated yeast) led to increased (66 mg l$^{-1}$ SCFAs compared to 57 mg l$^{-1}$ for the G1250S-M1250W mutated strain) and the insertion into the background of a R1834K-mutated strain to decreased SCFA production (37 mg l$^{-1}$ SCFAs compared to 100 mg l$^{-1}$ for R1834K; Fig. 2b). In spite of this partly ambiguous impact on overall FA amounts, specific influences of mutations on FA chain length can be derived from our data. Supplementary Table 2 gives an indication on the chain-length regulation effect contributed by a particular mutation.

Within tested strains, some combinations of mutations stand out: The highest total yield in extracellular SCFAs was achieved with the combination of I306A-R1834K-G1250S, producing total amounts of 118 mg l$^{-1}$. For the specific production of $C_6$-FA, the double mutant I306A-G1250S showed best results with 20 mg l$^{-1}$ of $C_6$-FA, accounting for 90% of the measured SCFAs ($C_6$ to $C_{12}$). Specific production of $C_8$-FA was possible with the triple mutant I306A-R1834K-F1279Y with a titre of 48 mg l$^{-1}$, representing 89% of its SCFA output.

**Vitality parameters of engineered strains.** For monitoring growth behaviour, we generally measured wet cell pellet weights

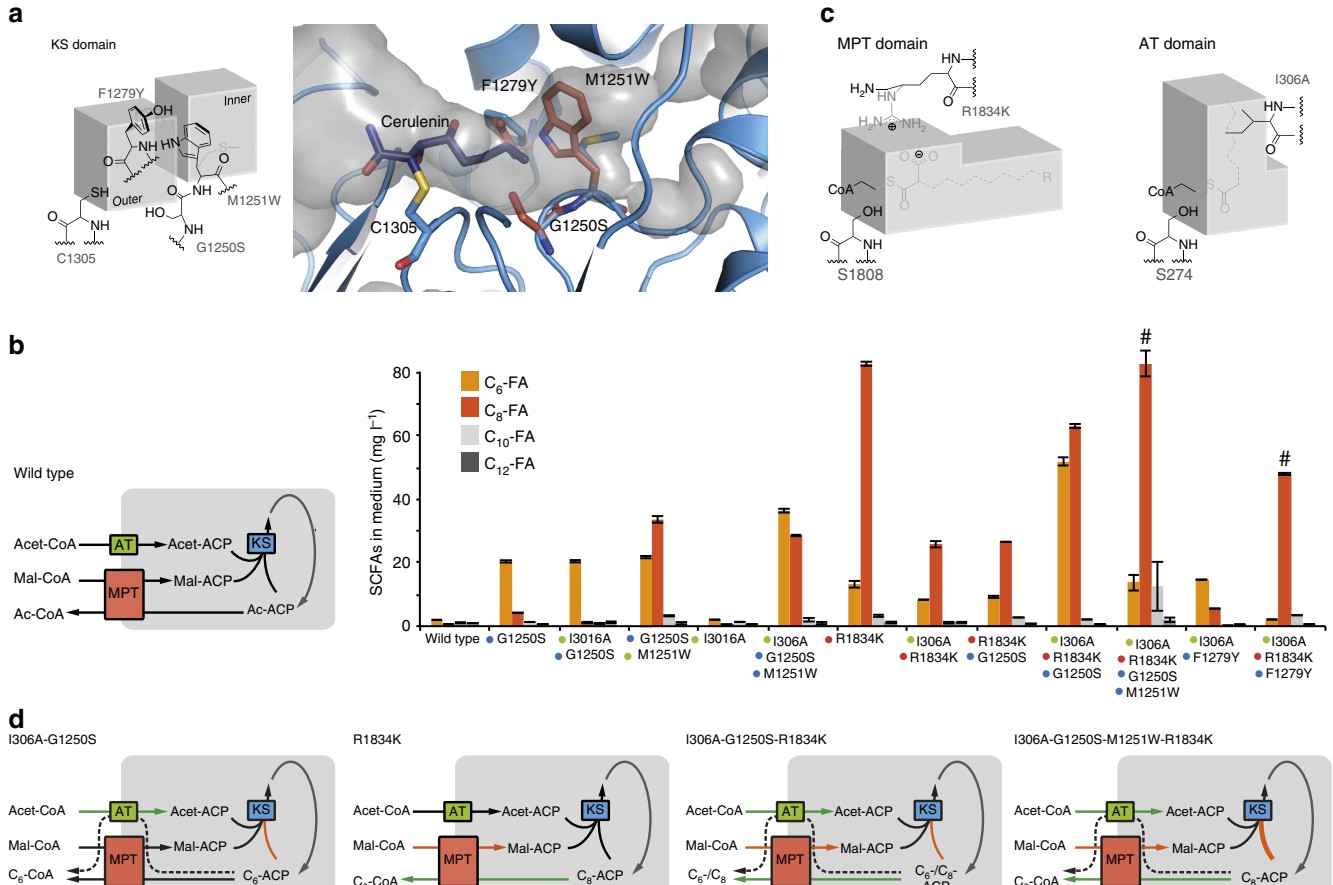

**Figure 2 | Active site engineering of S. cerevisiae FAS.** (**a**) KS active site mutated in three positions. (Right panel) X-ray structural data of KS with the binding channel in its open state (structure coloured in blue; PDB code 2VKZ (ref. 19)). Cerulenin, a FAS inhibitor and acyl mimic, which induces binding channel opening, is shown in dark blue. A structural model of G1250S-M1251W-F1279Y-mutated KS in its closed state is superimposed (residues shown in orange). (Left panel) Abstracted KS active site indicating an outer and an inner volume of the acyl binding channel, gated by methionine (native) or tryptophan (mutated FAS), respectively. Amino acids in grey show wild-type KS in its open-state conformation. Bound substrates are omitted for clarity. The closed-state conformation is shown for G1250S-M1251W-F1279Y-mutated KS (residues in black). (**b**) SCFA product spectra of selected mutated FAS strains in YPD medium. Error bars reflect the standard deviation from three technical replicates (collected on one biological sample). Biological repeatability is shown in Supplementary Fig. 1 and Supplementary Table 1. Please note, while **b** shows values from one biological sample, values in the text refer to means of two values (Supplementary Table 1). Mutations in KS are indicated by blue dots, and mutations in transferase domains MPT and AT by dots in red and green, respectively. Strains with I306A-R1834K-G1250S-M11251W-mutated FAS and I306A-R1834K-F1279Y-mutated FAS (both marked by #) grew only to approximately one-third of the regular cell density as compared to the other strains. A scheme on FAS-mediated FA synthesis is attached (flux scheme). Just enzymatic domains are shown that have been mutated in our approach (see for comparison Fig. 1b). (**c**) Cartoon representation of AT and MPT active sites. For MPT, the wild-type arginine is shown in grey. Bimodal binding is indicated by substrates with solid and dashed lines; acyl-CoA in dashed lines overlaid with malonyl-CoA in solid lines for MPT, and $C_8$-CoA (dashed) overlaid with acetyl-CoA (solid) for AT. (**d**) Flux schemes of engineered FASs estimated from data reported in literature, as well as deduced from the extracellular SCFA distribution. Red arrows indicate reduced and green arrows increased fluxes compared to wild type (**b**). A dashed line indicates a putatively new reaction channel installed by engineering. Flux schemes do not explain relative $C_6$-/$C_8$-FA levels.

as well as, for selected strains, recorded $OD_{600}$ at several time points (Supplementary Fig. 2 and Supplementary Table 3). In regular YPD, strains showed growth behaviours ranging from wild-type-like growth rates to extremely low or non-detectable growth rates. We assume that compromised growth relies on the insufficient supply with long FAs from *de novo* FA synthesis, as resulting from the efficient SCFA production. Supporting this hypothesis, growth defects can be compensated by addition of oleic acid ($C_{18:1}$, 1 mM) to the YPD medium (Supplementary Table 3). In YPD medium supplemented with oleic acid, extracellular FA concentrations were changed towards lower yields in SCFAs, with the only exception of the I306A-R1834K-G1250S-M1251W-mutated strain producing $131 \text{ mg l}^{-1}$ instead of $104 \text{ mg l}^{-1}$ of SCFAs (Supplementary Fig. 3). Intriguingly, we observed a shift towards shorter FA, as compared to growth in

regular YPD, with the G1250S-M1251W-mutated strain as most prominent example. It is tempting to speculate that this effect is based on an inhibition of acetyl-CoA carboxylase Acc1 (ref. 35), leading to lower malonyl-CoA levels and shorter FAs[33,34]. However, other more complex regulatory effects by long-chain FA, as for example on the FA cycle or on upstream enzymes, may be involved[35].

Regarding a putative impact on growth by toxic SCFAs[36,37], we were interested in clarifying whether a strong initial production of $C_8$-FA accounts for the inhibition of further growth. Product spectra of selected strains were recorded after 12, 24 and 48 h; the latter constituting the regularly chosen measurement point. While strains showed varying growth rates (Supplementary Fig. 2), the FA production profile was largely conserved reaching a maximum yield in SCFAs after 48 h (Supplementary Fig. 4). These data

argue against an early growth limitation of some strains by high inhibitory concentration of SCFAs. Glucose consumption and ethanol levels were recorded during the cultivations. All tested strains behaved similarly with glucose being entirely consumed after about 20 h, followed by a partial consumption of the produced ethanol (Supplementary Fig. 5).

**Effect of promoter exchange on FA synthesis**. For the production of SCFAs in *S. cerevisiae*, *FAS2* and *FAS1* were generally under control of their natural promoters and terminators[22]. We favoured this approach, as it is minimal-invasive and allows comparison with wild-type output spectra prior to applying means for FA overproduction[7]. To further evaluate the potential of our mutations, we performed a single layer of optimization, and probed FA production under FAS overexpression conditions. We therefore complemented the *Δfas1 Δfas2* strain with vectors, in which genes *FAS2* and *FAS1* are put under control of the alcohol dehydrogenase II promoter of *S. cerevisiae* (*pADH2*). We worked with pRS-*pADH2* vectors to generate R1834K-mutated (C$_8$-FA producing), I306A-G1250S-double mutated (C$_6$-FA) and I306A-G1250S-R1834K-triple mutated (C$_6$-FA and C$_8$-FA) yeast strains. As *pADH2* is repressed in the presence of glucose[38], and we have observed the consumption of glucose after about 20 h (Supplementary Fig. 5), we expected highest extracellular FA levels at longer cultivaton times. In addition, we buffered the medium by supplementing YPD medium with 250 mM potassium phosphate. Under these conditions, we observed a substantial increase in yields, with concentrations of C$_8$-FA of 245.0 mg l$^{-1}$ (R1834K; 85.9% of total extracellular SCFAs), C$_6$-FA of 71.5 mg l$^{-1}$ (I306A-G1250S; 78.2%), and mixed C$_6$-FA/C$_8$-FA of 464.4 mg l$^{-1}$ (I306A-G1250S-R1834K, 96.0%; Fig. 3a), which corresponds to the highest titre of SCFAs reported for yeast to date.

**The origin of short-chain acyl-CoA thioesterase activity**. Recently, Eht1 was described to act as octanoyl-CoA:ethanol acyltransferase and as thioesterase[39]. Assuming that the whole family of short-chain AEATases, comprising Eht1 (open reading frames YBR177C), Eeb1 (YPL095C) and Mgl2 (YMR210W), share this dual activity[40], we generated a set of strains deficient in candidate proteins to identify the origin of the hydrolysing activity, which processes short-chain acyl-CoAs to SCFAs that are finally exported to the culture medium. Specifically, we constructed strains with the single genes deleted (*Δeht1*, *Δeeb1* and *Δmgl2*), with deletions in two genes (*Δeeb1 Δeht1*, *Δeht1 Δmgl2*) and with all three genes deleted (*Δeeb1 Δeht1 Δmgl2*). All strains were also deficient in FAS (*Δfas1 Δfas2*). Working with this set of strains, complemented with engineered FAS, allowed addressing the question, whether the observed hydrolysis is accounted by a specific AEATase or by all three candidate AEATases.

The single deletions had a minor impact on extracellular SCFA levels, ruling out that a single AEATase is responsible for acyl-CoA hydrolysis. Data also demonstrate that upstream effects interfering in this study are unlikely (that is, deletions acting on FAS substrate levels, or deletions imposing indirect effects as for example regulatory effects). The two double-knockout strains generally caused reduced extracellular SCFA levels (particularly *Δeeb1 Δeht1*), while finally the triple knockout (*Δeeb1 Δeht1 Δmgl2*) led to essentially abolished production of SCFAs (Fig. 3b), suggesting that all three AEATases are responsible for short-chain acyl-CoA hydrolysis in SCFA producing strains.

Interestingly, AEATase deletions had different effects on FAS mutants. For both FAS mutants, R1834K and I306A-R1834K-G1250S, we found varying FA product distributions in AEATase single-knockout strains. In line with the reported specificity of Eht1 for C$_8$-CoA, we measured a significant drop of C$_8$-FA concentrations in the Eht1-deficient strain[39]. Similarly, our data suggest that Eeb1 has higher selectivity for C$_6$-CoA. Data of the double AEATase knockout strains support the specificities suggested from single knockouts (Fig. 3b). While the detailed molecular basis for the interplay of FAS and AEATases remains elusive, the observed phenomena are important *per se*. In our approach, which is powerful in producing SCFAs in narrow chain-length distributions, thioesterases (as AEATases) might impose a valuable downstream selectivity filter that can further improve purity and yield of free FA products.

## Discussion

The biosynthetic potential of nature for the processing of materials had been exploited for centuries, when the discipline biotechnology put an originally phenomenologically driven practice onto scientific grounds. Since then, the broad and efficient production of a large variety of compounds has been achieved, continuously replacing and complementing traditional synthetic chemical strategies. In spite of all progress, the accessible synthetic spectrum of products is still considerably restricted by the native function of the cellular components involved in the biosynthetic processes. Developing biotechnology to a versatile synthetic tool requires the modulation of specificities of natural systems to enlarge the spectrum of biotechnologically accessible compounds. Likewise, it is one of the foremost goals of bioengineering approaches, that aim at expanding the product portfolio, to conserve or even improve the intrinsic efficiency of natural systems for warranting high product yields[41].

In the presented study, we describe the minimally invasive engineering of *S. cerevisiae* FAS for the construction of SCFAs producing yeast strains. We identified a set of only five mutations that are capable of significantly controlling FA chain-length regulation. By applying mutations in several combinations, an overall titre of SCFAs of 464.4 mg l$^{-1}$ was achieved, which corresponds to a 109-fold increase over wild type, as well as the selective production of FAs of specific chain lengths (C$_6$-FA: 71.5 mg l$^{-1}$, 78.2% of total extracellular SCFAs; C$_8$-FA: 245.0 mg l$^{-1}$, 85.9%; Fig. 3a). The successful approach redirected kinetic fluxes of *de novo* FA synthesis by modulating substrate specificities of FAS enzymatic domains, while keeping the integrity of the protein intact. Mutations act directly on chain-length control, by interfering in the elongation of long acyl chains (KS mutations: G1250S, M1251W, F1279Y; Fig. 2a), as well as by favouring the release of short acyl chains through a weakened binding of the competing malonyl (MPT mutation R1834K) and a broadened acetyl binding channel (AT mutation I306A; Fig. 2c). In addition, mutations act indirectly on chain-length control by increasing acetyl/malonyl ratios, which pronounce priming of fatty acid synthesis and shift the spectrum towards shorter FA. By this approach, we intended to steer FA synthesis towards the premature release of the short acyl-chain without evoking a drop in synthetic rates. Among the mutations employed in FAS engineering, the MPT mutation is, however, clearly a costly one. It reduces the uptake of the malonyl moiety required for every new elongation cycle and likely affects turnover numbers. As a call for a first improvement of the presented system, FAS engineering needs to be optimized towards increased affinities of MPT for short-chain acyl-CoAs to increase acyl export rates without decreasing malonyl import.

Although specific information on the direct product spectra of FAS mutants is missing, we assume a bimodal chain-length

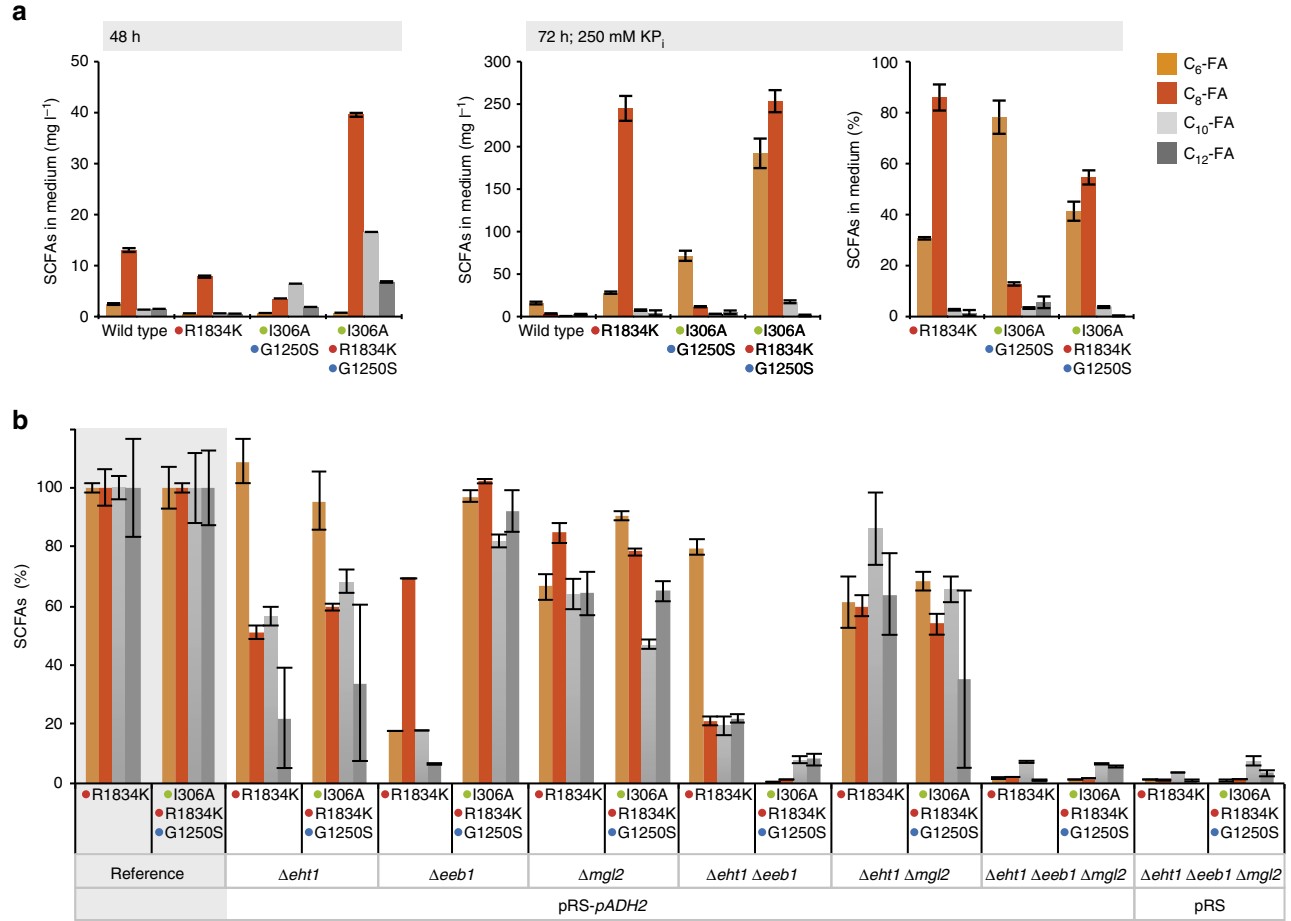

**Figure 3 | SCFA levels in FAS overproducing strains and AEATase knockout strains.** (**a**) Product spectra of *S. cerevisiae Δfas1 Δfas2* complemented with selected FAS expressed from *pADH2* promoter-controlled genes. Expression conditions as described in Fig. 2b, with cultures grown for 48 h in non-buffered and 72 h in buffered medium (250 mM potassium phosphate, KPi). Experiments were performed with statistics as described in Fig. 2b. For cultures grown for 72 h, product distribution is also given in shares of total yields. (**b**) Percentage of extracellular FAs detected in medium in AEATase knockout strains. All AEATase knockout strains are also deficient in FAS (*Δfas1 Δfas2*; not indicated by figure labels). Strains are complemented with SCFA producing FAS mutants, as indicated, with genes under control of the wild type and the *pADH2* promoter. Recorded extracellular SCFA concentrations are given in percentage and relate to reference (*Δfas1 Δfas2*), which is set to 100% (grey background). Cultivation was performed for 72 h in 100 mM buffered phosphate medium. Experiments were performed with statistics as described in Fig. 2b.

distribution of products composed of short-chain acyl-CoAs, which are finally exported as free FAs to the medium, and normal length $C_{16}$- and $C_{18}$-CoA. This assumption is based on data that were collected on cerulenin derivatives modulated in the length of their aliphatic moiety (for cerulenin binding to KS, see Fig. 2a). These derivatives showed a bimodal profile in inhibition strength, indicating weak binding of compounds of medium length[42]. A bimodal product spectrum of FAS variants may be explained in a similar manner as originating from KS engineering. Short-chain acyl-CoA esters are produced, as the respective acyl chains are hindered in KS binding by clashing with the gate keeper residue. Long-chain acyl-CoA esters result from acyl chains that escape early release and grow into the inner volume of the KS-binding channel, where they finally underlie constraints of original chain-length control (Fig. 2a). Such a leaky chain-length control can explain the viability of most of the FAS-mutated strains, as well as the growth deficit in FAS variants with accumulated critical mutations (see Supplementary Fig. 2 and Supplementary Table 3).

In our study, we have further identified three AEATases (Eeb1, Eht1 and Mgl2) that are responsible for the hydrolysis of short-chain acyl-CoA esters[39,40]. In a combinatorial approach, generating a set of AEATase-deficient strains, we were able to reveal the collaboration of all three proteins for hydrolysis (Fig. 3b). AEATases, as well as thioesterases in general, can be valuable tools when assisting *de novo* chain-length control. For example, for the putative task of the specific production of $C_8$-FA, the Eeb1-deficient strain might be the better suited host for R1834K-mutated FAS than the wild-type strain, as it leads to higher product purity (Fig. 3b). In such combined approaches, recombinant thioesterases with narrow product spectra might be even more powerful[23]. AEATases and thioesterases can also be useful in an inverse setting. Short-chain acyl-CoAs produced with the presented FAS mutants might be ultra-specifically directed into downstream biosynthetic pathways processes, when clearing the cytoplasm from unwanted acyl-CoAs by selective thioesterase activity[4,43,44].

Our study on structure-based rational mutagenesis of *S. cerevisiae* FAS represents a novel minimally invasive approach for chain-length control in *de novo* FA synthesis. We succeeded in generating a set of mutations that is capable of inducing SCFA production by redirecting kinetic fluxes in *de novo* FA synthesis by FAS. The presented system can be seen as a module that can turn on SCFA production in essentially any microbial organism. With the advantage of gaining access

to CoA esters, as the immediate products of fungal FA synthesis, this technology is also highly attractive for biosynthetic processes that aim at the synthesis of FA derived substances, for example, as alkanes and alcohols for biofuel production[4,45], or at providing acyl-CoAs as starter compound for other biosynthetic clusters[43].

## Methods

**Description yeast strain.** The haploid *S. cerevisiae* strain BY.PK1238_1A_KO, which was kindly provided by Peter Kötter (University of Frankfurt, Germany) and used in this work, has a BY background and the reading frames of *FAS1* and *FAS2* are each replaced by a kanMX4 cassette, resulting in a clean knockout of FAS and antibiotic resistance against geneticin. The exact genotype is *Matα; ura3Δ0; his3Δ0; leu2Δ0; TRP1; lys2Δ0; MET15; fas1::uptag-kanMX4-downtag; fas2::uptag-kanMX4-downtag*. The deletion of three thioesterase genes (*EEB1*, *EHT1* and *MGL2*) was performed using the simplified CRISPR-Cas genome editing tool for *S. cerevisiae*[46].

**Vector description.** The vectors used in this work are centromeric pRS shuttle vectors of types pRS313 and pRS315 (ref. 47) with single copy number and HIS3 and LEU2 auxotrophy marker, respectively. *FAS1* or mutations thereof were always provided on pRS315, while *FAS2* or mutations thereof were always provided on pRS313, each regulated by its corresponding native promoter (995 bp upstream for *FAS1* and 480 bp upstream for *FAS2*)[22]. Terminator sequences were set to 295 bp and 258 bp, respectively, downstream of the open reading frames. Cloning was always done in *E. coli* using the Infusion HD cloning kit (Clontech, Mountain View, USA). Wild-type *FAS1* and *FAS2* genes were assembled from several fragments, which were amplified from *S. cerevisiae* genomic DNA, into pRS vectors using BamHI and SalI restriction sites. Exact chromosomal coordinates including promoter and terminator sequences according to strain S288C are for *FAS1* (YKL182w): Chr XI 99676-107121 and for *FAS2* (YPL231w): ChrXVI 108172-114573. In addition to pRS-based expression, FAS encoding genes *FAS1* and *FAS2*, in wild-type and mutated variants, were also provided under control of the *S. cerevisiae* alcohol dehydrogenase 2 promoter (ADH2; 573 bp). Vectors are similar to pRS-based vectors, described above, except that the promoter has been exchanged. Vectors are termed pRS-pADH2. Primers for site-directed mutagenesis are listed in Supplementary Table 4.

**Transformation.** For yeast transformations, ∼1 μg of each plasmid DNA was co-transformed following a modified lithium acetate protocol[48]. A 5 ml overnight culture of strain BY.PK1238_1A_KO in YPD (1% yeast extract, 2% peptone, both produced by BD, Difco Laboratories, Sparks, USA; 2% dextrose, purchased from Roth, Karlsruhe, Germany) containing 200 μg ml$^{-1}$ geneticin disulfate, free FAs (myristic, palmitic and stearic acid, each 50 μg ml$^{-1}$) and 1% Tween20 grown at 30 °C and 200 r.p.m. was used to inoculate a main culture in the same medium. After shaking at 30 °C and 200 r.p.m. to OD$_{600}$ = 0.8, a volume of 5 ml of this culture was harvested by centrifugation (3,000 r.c.f., 5 min, 24 °C). The cells were washed by resuspending in 1 ml water and centrifuged again. After resuspension in lithium acetate solution (0.1 M), cells were incubated for 5 min at 24 °C and centrifuged (5,000 r.c.f., 15 s, 24 °C), before the transformation mix was added (240 μl PEG 1,500 solution (50%), 76 μl water, 36 μl lithium acetate solution (1.0 M), 5.0 μl single-stranded DNA solution from salmon testis (10 mg ml$^{-1}$), 2 μl of each plasmid DNA solution). The cell suspension was mixed well and incubated for 30 min at 30 °C followed by 20 min at 42 °C. After pelleting the cells by centrifugation (4,000 r.c.f., 15 s, 4 °C), they were washed with 1 ml water, pelleted again (4,000 r.c.f., 15 s, 4 °C) and resuspended in 100 μl water. For selection of the yeast transformants, the cell suspension was spread on SCD–his–leu agar plates containing 200 μg ml$^{-1}$ geneticin disulfate, free FAs (myristic, palmitic and stearic acid, each 50 μg ml$^{-1}$) and 1% Tween20.

**Cultures for product analysis.** *S. cerevisiae* strains were cultured in YPD medium (composition as described above). For cultivation of FAS encoded by genes under the *pADH2* promoter for 72 and 96 h, the medium was additionally buffered with 100 or 250 mM potassium phosphate (as indicated in figure legends). For product analyses, several colonies of the *S. cerevisiae* strains were picked and combined in one pre-culture (5 ml YPD with 200 μg ml$^{-1}$ geneticin disulfate, 50 mg ml$^{-1}$ final concentration). After shaking at 200 r.p.m. at 30 °C over night, the OD$_{600}$ was measured. The main culture (50 ml YPD with 200 μg ml$^{-1}$ geneticin disulfate, 50 mg ml$^{-1}$ final concentration) was inoculated to OD$_{600}$ = 0.1 and shaken for 48 h at 200 r.p.m. and 30 °C. Before further processing, the OD$_{600}$ was measured again. For samples with long FAs supplementation, C$_{18:1}$ and Tergitol NP-40 (solution in water, 70%) were added to all cultures to a final concentration of 1 mM or 1% in the case of Tergitol. Buffered medium was used for strains with FAS encoding genes under the pADH2 promoter.

**Sample processing.** FA extraction was performed as follows[23]: first cells were spun down at 3,500 r.c.f. for 15 min. The supernatant was aliquoted in 10 ml portions and 0.2 mg of the internal standard, heptanoic acid (C$_7$), dissolved in

100 μl of solvent (33% methanol, 67% chloroform) was added. After acidification with 1 ml HCl (1 M), 2.5 ml of a mixture of equal amounts of methanol and chloroform were added. The samples were shaken vigorously for 5 min and then centrifuged again at 3,500 r.c.f. for 10 min. The chloroform layer was transferred to a new vial and any residual water removed. The liquid was then fully evaporated in a SpeedVac. For methylation, 0.3 ml of HCl (8% solution made from 9.7 ml of concentrated HCl and 41.5 ml of methanol), 0.2 ml of toluene and 1.5 ml of methanol were added for resuspension[49]. The mixture was vortexed and later heated to 100 °C for 3 h. After a cool down on ice, 1 ml hexane and 1 ml water were added. The samples were vortexed once again and after separation, the organic layer was transferred in a vial for further analysis by gas chromatography (GC).

**Determination of free FAs by GC.** The resulting FA methyl esters (dissolved in hexane) were measured with a Perkin Elmer Clarus 400 gas chromatograph (Perkin Elmer, Rodgau, Germany) equipped with an Elite FFAP capillary column (30 m × 0.25 mm, film thickness: 0.25 μm; Perkin Elmer, Rodgau, Germany) and a flame ionization detector (Perkin Elmer, Rodgau, Germany). A volume of 1 μl of the sample was analysed after split injection (10 ml min$^{-1}$) and helium as carrier gas. The temperatures of the injector and detector were 200 and 250 °C, respectively. The following temperature program was applied: 50 °C for 5 min, increase of 10 °C min$^{-1}$ to 120 °C (hold for 5 min), increase of 15 °C min$^{-1}$ to 180 °C (hold for 10 min), and increase of 20 °C min$^{-1}$ to 220 °C for 37 min.

For each culture, three 10 ml aliquots from the same media were separately processed and measured by GC. The standard deviation between the three measurements from the same media was typically below 1 mg l$^{-1}$ (for 92% of the measured concentrations).

**Metabolite analysis by HPLC.** For quantification of glucose and ethanol, 450 μl cell-free samples were mixed with 50 μl of 50% (w/v) 5-sulfosalicylic acid, vigorously shaken and centrifuged (4 °C, 5 min, 13,000 r.c.f.). The supernatant was analysed with an UHPLC+ system (Dionex UltiMate 3000, Thermo Scientific, USA) equipped with a HyperREZ XP Carbohydrate H+ 8 μm column. To detect the substrates, a refractive index detector (Thermo Shodex RI-101) was used. Separation was carried out at 65 °C with 5 mM sulfuric acid as mobile phase (flow rate of 0.6 ml min$^{-1}$). Five standards (mixtures of D-glucose, ethanol, glycerol and acetate with concentrations of 0.05–2% (w/v)) were analysed for quantification of the different compounds.

**Determination of cell density.** The cell density in a liquid culture was measured with an Ultrospec 2100 pro spectrophotometer (GE Healthcare, USA) by determination of the optical density at 600 nm (OD$_{600}$).

**Data availability.** All data generated or analysed during this study are included in this published article (and its Supplementary Information file) or are available from the corresponding author upon reasonable request.

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

## Acknowledgements

We thank Dieter Oesterhelt (Martinsried, Germany) and Floris P. Buelens (Göttingen, Germany) for valuable discussions on chain-length control and FAS catalytic mechanisms. We are further grateful to Mislav Oreb (Frankfurt, Germany) for helpful advice, and Peter Kötter (Frankfurt, Germany) for providing the double-knockout strain. Work in the lab of E.B. was supported by the Hessian Ministry of Science and Art via the LOEWE research focus 'Integrative fungal research (IPF)'. This work was also supported by a Lichtenberg Grant of the Volkswagen Foundation to M.G. (grant number 85701).

## Author contributions

J.G. and R.P. performed biological studies; J.G., R.P., E.B. and M.G. designed research and analysed data; M.F. performed initial cloning and transformations; J.G., R.P., E.B. and M.G. wrote the manuscript.

## Additional information

**Competing financial interests:** J.G., R.P., E.B. and M.G. are inventors of EP patent application 15 162 192.7 filed on 1 April 2015, and J.G., E.B. and M.G. are inventors of EP patent application 15 174 342.4 filed on 26 June 2015, by Goethe-University Frankfurt.

