## [Peer Review File · Nature Communications]

Reviewers' comments:

Reviewer #1, expert in yeast metabolic engineering and biofuels (Remarks to the Author):

In the manuscript by Gajewski et al., the authors engineered a yeast FAS to produce short-chain (C6-C8) instead of long-chain (>C12) fatty acids. The results of this study are solid but not necessarily interesting. If this manuscript aims to demonstrate that the engineered FAS could be a promising route for producing short-chain fatty acids at high yield, then pathway engineering as well as detailed sequencing and flux analysis need to be presented, and more importantly, the titer of C6- and C8-fatty acid needs to be dramatically improved compared to current value (<120 mg/L). Also, this manuscript lacks the justification of using FAS to produce short-chain fatty acids. There are other approaches such as introducing heterologous FAS (PMID: 23928901) and enzyme compartmentalization (PMID: 27621436) to shorten the chain length of fatty acids. Therefore, the approach presented in this manuscript is not completely novel in terms of making short-chain fatty acids. Overall, although the importance of making short-chain fatty acids is well recognized, the discovery presented in this manuscript is not significant enough to be published in Nature Communications.

Reviewer #2, expert in bioengineering and FAS (Remarks to the Author):

The article "Engineering fundal de novo fatty acid synthesis for short fatty acid production" by Gajewski, et al., addresses the modification of the FAS in yeast for the production of short fatty acids. The group makes use of a number of sites for mutagenesis, many of them already studied in isolation, the combination of which provide unique access to C6 and C8 fatty acid analogs. Here the short chain fatty acids are excreted into the media, and the authors demonstrate production of 118 mg/L. This manuscript displays the strengths of this research group to analyze and interpret the intricacies of large structures. Important to successful analysis this work was the preparation of FAS null mutants such that the site-directed FAS genes can be added over a "blank" expression background. Finally, the study of thioesterase knockouts convincingly provides a rationale for release and excretion of short fatty acids. In conclusion, this study is very well conceived and concisely explained. What is lacking in novelty is made up for by careful analysis and interpretation.

The following points should be addressed:

1. The G1250S strain is described in the main text to provide 15.3 mg/L of C6, but this does not agree with the histogram in Fig 2.
2. What is not addressed in the manuscript is the affect of these FAS mutations, and modified fatty acid production, on cell morphology and robustness.
3. The FAS null strain is incorrectly called $\Delta fas1/ \Delta fas1$ in the manuscript.

Reviewers' comments:

Reviewer #1, expert in yeast metabolic engineering and biofuels (Remarks to the Author):

In the manuscript by Gajewski et al., the authors engineered a yeast FAS to produce short-chain (C6-C8) instead of long-chain (>C12) fatty acids. The results of this study are solid but not necessarily interesting. If this manuscript aims to demonstrate that the engineered FAS could be a promising route for producing short-chain fatty acids at high yield, then pathway engineering as well as detailed sequencing and flux analysis need to be presented, and more importantly, the titer of C6- and C8-fatty acid needs to be dramatically improved compared to current value (<120 mg/L). Also, this manuscript lacks the justification of using FAS to produce short-chain fatty acids. There are other approaches such as introducing heterologous FAS (PMID: 23928901) and enzyme compartmentalization (PMID: 27621436) to shorten the chain length of fatty acids. Therefore, the approach presented in this manuscript is not completely novel in terms of making short-chain fatty acids. Overall, although the importance of making short-chain fatty acids is well recognized, the discovery presented in this manuscript is not significant enough to be published in Nature Communications.

Reply

As mentioned by Reviewer #1, microbial production of short chain fatty acids (SCFAs) has been reported before. We are referring to this work in the manuscript. The paper of Stephanopoulos and coworkers (PMID: 27621436) has been published after manuscript submission, but is now also included in the manuscript. In the following, we try to convince that our approach of SCFA production is novel, in spite of reported previous achievements, and warrants publication in Nature Communication.

First and foremost, we would like to emphasize that it was not our intention to engineer yeast strains for the production of short-chain fatty acids in industrially reliable yields, but to provide a completely new concept for their production. Our concept focuses on the direct engineering of fatty acid synthase (FAS)-mediated chain length control – different to all previous approaches, which are all based on short chain acyl-ACP thioesterases (TEs). Nevertheless, even without any pathway engineering, SCFA production with our approach already provides titers exceeding those of all previous approaches demonstrating the high potential of our concept. For being more specific in our reply, we would like to focus on three points:

(1) Fungal FASs are the best-suited systems for FA production: Fungal FASs are ideal objects for engineering microbial FA synthesis for biotechnological purposes. As calculated from the reported specific activity of 2500 mU/mg, *S. cerevisiae* FAS runs 18.2 iterative cycles per second (per set of active sites), which is roughly 5 times faster than any other FAS multienzyme. Given that each cycle requires six productive interactions between the substrate shuttling ACP and the catalytic domain (ACP:KS (ping) → ACP:MTP → ACP:KS (pong) → ACP:KR → ACP:DH → ACP:ER), *S. cerevisiae* FAS performs a catalytic step every 0.9 msec. This tremendous catalytic efficiency is based on the highly evolutionarily developed architecture of fungal FASs. Enzymatic domains are rigidly embedded into the walls of reactions chambers, and substrate shuttling by the acyl carrier protein (ACP) is subtly balanced between electrostatic steering and molecular crowding properties. Several studies have recently described the mechanistic basis of substrate shuttling¹⁻³.

(2) Efficient engineering strategies require sustainable protein design: Current strategies in producing SCFAs have in common to impose chain length control by hijacking native synthesis with a short chain acyl-ACP specific TEs. These set-ups either work with (i) the fungal FASs or the less efficient type II FAS system by in both cases co-expressing the short chain acyl-ACP specific TE^{4,5}, (ii) the less efficient mammalian FAS by exchanging the native with the short chain acyl-ACP specific TE⁶, or (iii) the fungal FAS by exchanging the MPT with short chain acyl-ACP specific TEs⁵. (We note that the protein design by Stephanopoulos and coworkers has to be regarded as “violating” fungal FAS properties. First, the inserted TE domain does not provide the interface for direct interactions with the fungal ACP and, therefore, is not or just poorly able to participate in substrate shuttling. Second, the domain exchange affects the function of malonyl loading, which is provided by the MPT domain.)

The novelty of the presented approach lies in our minimal invasive fungal FAS design for SCFAs production. By solely modifying substrate specificities, we intended to steer *de novo* fatty acid synthesis towards the early release of not yet fully elongated C₁₆ and C₁₈-CoA, while essentially leaving the overall molecular mechanisms of the protein unchanged. This approach is powerful. First, we work with fungal FASs, which are, owing to their intrinsic catalytic properties, the superior “nano-compartment” for microbial FA synthesis, and, second, we maintain the high synthetic rate of fungal FASs by the minimal invasive design. According to the high SCFAs titers, this engineering strategy is indeed successful.

There is another important advantage connected to the described engineering approach. As we are manipulating the inherent chain length control of FASs (i.e. different to all other approaches, we do not work with TEs that hydrolyze short chain acyl-ACP), we receive short-chain acyl-CoA esters as immediate products. This is highly advantageous, when the desired products are not just SCFAs, but also derivatives as aldehydes, alcohols and alkanes. The short-chain acyl-CoA esters, received from our manipulated fungal FA synthesis, can directly be processed in further reactions. This is different to TE-based chain length control, which requires reactivation of the free FA to the acyl-CoA ester by an ATP-consuming fatty acyl-CoA synthetase (FadD) (see e.g. ref ⁴). Any pathway that can accept acyl-CoA esters is potentially compatible with our FAS design. In a systematic study, we have identified three AEATases as the origin of native TE activity. AEATase knockouts strains prevent sequestering of SCFAs. These are interesting strains as they increase the lifetime of short chain acyl-CoA esters and putatively supporting downstream processing.

(3) Evaluation of the potential of the minimal invasive protein design: Encouraged by the Reviewer's comment, we have performed new experiments, and put FAS encoding genes under the control of the *S. cerevisiae* alcohol dehydrogenase 2 (ADH2) promoter. Since this experimental set-up still does not include pathway engineering, as e.g. such powerful strategies as blocking β -oxidation ⁷, these experiments are informative for evaluating the potential of our minimal invasive FAS constructs. We received extracellular titers of SCFA (mix of C₆- and C₈-FA) of up to 464 mg/L, and additionally achieved the specific production of C₆-FA and of C₈-FA in yields of 72 mg/L (78% of total SCFAs) and 245 mg/L (86%), respectively. These titers are significantly exceeding the values reported to date ⁶, and clearly underline the success of our protein design. To the best of our knowledge, the specific production of FA has not been reported before. While, clearly, the presented technology will not bridge the gap from an academic exercise to a real industrial application of microbial SCFA production, it should be considered as an important step in a long-term development. Particularly, as our approach in protein engineering is sustainable in the light of catalytic efficiency and energy conservation (both properties largely decide on the industrial relevance of such a technology), we consider our study as an important development to the field.

Reviewer #2, expert in bioengineering and FAS (Remarks to the Author):

The article "Engineering fundal de novo fatty acid synthesis for short fatty acid production" by Gajewski, et al., addresses the modification of the FAS in yeast for the production of short fatty acids. The group makes use of a number of sites for mutagenesis, many of them already studied in isolation, the combination of which provide unique access to C6 and C8 fatty acid analogs. Here the short chain fatty acids are excreted into the media, and the authors demonstrate production of 118 mg/L. This manuscript displays the strengths of this research group to analyze and interpret the intricacies of large structures. Important to successful analysis this work was the preparation of FAS null mutants such that the site-directed FAS genes can be added over a "blank" expression background. Finally, the study of thioesterase knockouts convincingly provides a rationale for release and excretion of short fatty acids. In conclusion, this study is very well conceived and concisely explained. What is lacking in novelty is made up for by careful analysis and interpretation.

The following points should be addressed:

- 1. The G1250S strain is described in the main text to provide 15.3 mg/L of C6, but this does not agree with the histogram in Fig 2.*
- 2. What is not addressed in the manuscript is the affect of these FAS mutations, and modified fatty acid production, on cell morphology and robustness.*
- 3. The FAS null strain is incorrectly called $\Delta fas1/ \Delta fas1$ in the manuscript.*

Reply

For underlining the novelty of our approach, we would like to refer to our reply to the comments of Reviewer 1. Specific points raised by Reviewer 2:

- (1) We are very grateful for the careful inspection of the manuscript, even down to the details of comparing data in text and in figures. This revealed an inconsistency in the presentation of data, as Fig. 2B shows data of one independent experiment (one biological sample measured in technical replicates), whereas we refer to the mean of two independent experiments in the text (as well as in Supplementary Table 2). We now show data of both experiments in the new Supplementary Table 1. Additionally, we explain our experimental set-up in more detail.
- (2) We have not performed experiments addressing cell morphology. The robustness of the cells has been addressed, however, as part of our studies towards growth behavior (growth curve and cell wet weight), and glucose and ethanol profiles (in reference to wild type yeast). From these data, we can say that strains with accumulated FAS mutations are rather affected in growth, and we interpret

these data on the basis of a bimodal distribution of acyl-CoA produced by these FASs. Accordingly, FAS variants that are efficient in short chain acyl-CoA production might tend to insufficiently cover the basal cellular need of long chain acyl-CoA, which affects growth.

(3) Corrected in the revised manuscript.

Overview of changes:

We have made the following changes:

(1) Editing: Although we present the new data, the manuscript is shortened by almost 10%. We shortened the plain description of the structural properties of fungal FAS, to, instead, disclose in more detail the highly optimized synthetic concept of the protein. We particularly do this in the introduction section. Specifically, we compare fungal FAS with other type I FAS systems, and explain the important concept of substrate shuttling. This section should raise the awareness of the reader towards the importance of an engineering concept that preserves the molecular concepts of fungal FASs (and other proteins). In addition to the introduction section, we have also edited the discussion section. We more strongly pronounce the dual benefit of following such a minimal invasive engineering approach: (i) Judged by the high titers of SCFAs found in the culture medium and the access to SCFAs of specific chain lengths, the approach proves to be indeed efficient. (ii) The direct manipulation of chain length control delivers short acyl-CoA esters, which can be directly processed in downstream pathways, and gives direct access to a broad spectrum of compounds.

(2) New data: We have included new data collected on FAS encoded by genes under control of the stronger ADH2 promoter. Data are presented in the new paragraph "Effect of promoter exchange on FA synthesis" and in Fig 3A. Experimental procedures, materials and methods for collecting the new data have been included in the revised manuscript. We have further moved the previous Fig. 3A, showing the time course of FA production, to the Supplementary Information. Finally, we have added Supplementary Table 1, in which we present data of the two independent experiments.

(3) Additional changes: We have updated references, i.e. including the recent work of Stephanopoulos and coworkers. Finally, we have also made some minor changes, e.g. changing our nomenclature towards "short chain" instead of solely "short" (see also change in title), and introduced the abbreviation SCFA for short chain fatty acid.

- 1 Anselmi, C., Grininger, M., Gipson, P. & Faraldo-Gomez, J. Mechanism of substrate shuttling by the acyl-carrier protein within the fatty acid mega-synthase. *J Am Chem Soc* **132**, 12357-12364 (2010).
- 2 Gipson, P. *et al.* Direct structural insight into the substrate-shuttling mechanism of yeast fatty acid synthase by electron cryomicroscopy. *Proc Natl Acad Sci U S A*. **107**, 9164-9169 (2010).
- 3 Beld, J., Lee, D. J. & Burkart, M. D. Fatty acid biosynthesis revisited: structure elucidation and metabolic engineering. *Mol Biosyst* **11**, 38-59 (2015).
- 4 Choi, Y. J. & Lee, S. Y. Microbial production of short-chain alkanes. *Nature* **502**, 571-574 (2013).
- 5 Xu, P., Qiao, K., Ahn, W. S. & Stephanopoulos, G. Engineering *Yarrowia lipolytica* as a platform for synthesis of drop-in transportation fuels and oleochemicals. *Proc Natl Acad Sci U S A* **113**, 10848-10853 (2016).
- 6 Leber, C. & Da Silva, N. A. Engineering of *Saccharomyces cerevisiae* for the synthesis of short chain fatty acids. *Biotechnol Bioeng*. **111**, 347-358 (2014).
- 7 Zhou, Y. J. *et al.* Production of fatty acid-derived oleochemicals and biofuels by synthetic yeast cell factories. *Nature Commun* **7**, 11709 (2016).

REVIEWERS' COMMENTS:

Reviewer #2 (Remarks to the Author):

The authors have addressed all reviewer criticisms appropriately.